Identification of key microRNAs and the underlying molecular mechanism in spinal cord ischemia-reperfusion injury in rats

Chen Fengshou
Han Jie
Wang Dan wonder12251@hotmail.com
Department of Anesthesiology, the First Hospital of China Medical University , Shenyang, Liaoning province , China
Linden Rafael
Electronic publication date: 2021 May 27
Publication date: 2021
Volume: 9
Electronic Location ID: e11454
Received 2020 Nov 17; Accepted 2021 Apr 23
Copyright: © 2021 Chen et al.
Copyright year: 2021
Copyright holder: Chen et al.
License: This is an open access article distributed under the terms of the Creative Commons Attribution License, which permits unrestricted use, distribution, reproduction and adaptation in any medium and for any purpose provided that it is properly attributed. For attribution, the original author(s), title, publication source (PeerJ) and either DOI or URL of the article must be cited.
License URL: https://creativecommons.org/licenses/by/4.0/

Keywords: microRNA, Spinal cord ischemia reperfusion injury, Bioinformatics analysis, Transcription factor analysis

Funding: The authors received no funding for this work.

==============================
Spinal cord ischemia-reperfusion injury (SCII) is a pathological process with severe complications such as paraplegia and paralysis. Aberrant miRNA expression is involved in the development of SCII. Differences in the experimenters, filtering conditions, control selection, and sequencing platform may lead to different miRNA expression results. This study systematically analyzes the available SCII miRNA expression data to explore the key differently expressed miRNAs (DEmiRNAs) and the underlying molecular mechanism in SCII. A systematic bioinformatics analysis was performed on 23 representative rat SCII miRNA datasets from PubMed. The target genes of key DEmiRNAs were predicted on miRDB. The DAVID and TFactS databases were utilized for functional enrichment and transcription factor binding analyses. In this study, 19 key DEmiRNAs involved in SCII were identified, 9 of which were upregulated (miR-144-3p, miR-3568, miR-204, miR-30c, miR-34c-3p, miR-155-3p, miR-200b, miR-463, and miR-760-5p) and 10 downregulated (miR-28-5p, miR-21-5p, miR-702-3p, miR-291a-3p, miR-199a-3p, miR-352, miR-743b-3p, miR-125b-2-3p, miR-129-1-3p, and miR-136). KEGG enrichment analysis on the target genes of the upregulated DEmiRNAs revealed that the involved pathways were mainly the cGMP-PKG and cAMP signaling pathways. KEGG enrichment analysis on the target genes of the downregulated DEmiRNAs revealed that the involved pathways were mainly the Chemokine and MAPK signaling pathways. GO enrichment analysis indicated that the target genes of the upregulated DEmiRNAs were markedly enriched in biological processes such as brain development and the positive regulation of transcription from RNA polymerase II promoter. Target genes of the downregulated DEmiRNAs were mainly enriched in biological processes such as intracellular signal transduction and negative regulation of cell proliferation. According to the transcription factor analysis, the four transcription factors, including SP1, GLI1, GLI2, and FOXO3, had important regulatory effects on the target genes of the key DEmiRNAs. Among the upregulated DEmiRNAs, miR-3568 was especially interesting. While SCII causes severe neurological deficits of lower extremities, the anti-miRNA oligonucleotides (AMOs) of miR-3568 improve neurological function. Cleaved caspase-3 and Bax was markedly upregulated in SCII comparing to the sham group, and miR-3568 AMO reduced the upregulation. Bcl-2 expression levels showed a opposite trend as cleaved caspase-3. The expression of GATA6, GATA4, and RBPJ decreased after SCII and miR-3568 AMO attenuated this upregulation. In conclusion, 19 significant DEmiRNAs in the pathogenesis of SCII were identified, and the underlying molecular mechanisms were validated. The DEmiRNAs could serve as potential intervention targets for SCII. Moreover, inhibition of miR-3568 preserved hind limb function after SCII by reducing apoptosis, possibly through regulating GATA6, GATA4, and RBPJ in SCII.

Introduction

Spinal cord ischemia-reperfusion injury (SCII) is the damage caused by the restoration of blood perfusion in the ischemic spinal cord tissues (Fang et al., 2015). The normal function of the spinal cord may be hindered by SCII, resulting in severe complications such as paraplegia and paralysis (Smith et al., 2012). Current drugs and therapeutic measures for SCII are effectual but deficient. Hence, SCII remains a significant challenge in clinical therapy (Xu et al., 2014a). It is necessary to identify new molecular targets for SCII.

MicroRNAs (miRNAs) are non-coding RNAs 20 to 22 nt in length that regulate target genes expression at the post-transcriptional level via base-pairing with the 3′-untranslated region (3′-UTR) of mRNA (Balsam, 2017; Zhai et al., 2012). Due to the involvement in the numerous biological processes, such as neurogenesis, inflammation, apoptosis, and autophagy, miRNAs may conduce to the pathogenesis of the central nervous system (CNS) disorders, including SCII, cerebral ischemia-reperfusion injury (CIRI), spinal cord injury (SCI), and Parkinson’s disease (Bhalala, Srikanth & Kessler, 2013; Li et al., 2018a; Li et al., 2018b; Wang et al., 2020b). Moreover, aberrant miRNAs expression has been connected to the development of SCII at different stages (Hu, Lv & Yin, 2013; Li et al., 2016b; Liu et al., 2020).

This study aims to furnish the authentic miRNA data and predict the target genes linked with the occurrence and development of SCII. The role of the contrastingly expressed miRNAs in regulating the target genes during SCII and the altered miRNA-TF regulatory patterns were established based on the TFactS database (http://www.tfacts.org/) to provide significant clues for targeting the key miRNAs as molecular markers in the treatment of SCII. The involvement of selected miRNA and its target genes in SCII were also verified.

Materials and Methods

Identification of the key differentially expressed miRNAs in SCII

The existing studies on the miRNA expressions in SCII were located on PubMed with keywords such as miRNA and spinal cord ischemic reperfusion injury on 30 November 2020. The species involved in the related studies were limited to rats. The information on SCII and the miRNA sequencing samples and the detection methods for the differentially expressed miRNAs (DEmiRNAs) were extracted from the relevant studies. The upregulated and downregulated DEmiRNAs in SCII compared with the sham groups were also extracted. The DEmiRNAs occurrence of each dataset was calculated, and DEmiRNAs that appeared in at least two datasets were defined as key DEmiRNAs.

Predicting the target genes of DEmiRNAs in SCII

The target genes of DEmiRNAs were predicted on miDRB with the gene target score set to over 80.

Gene ontology and KEGG enrichment analysis on the target genes of DEmiRNAs in SCII

Information on the target genes of DEmiRNAs was obtained from the DAVID database for KEGG pathway analysis and GO enrichment analysis (Guo et al., 2019). The cut-off criteria for the Kyoto Encyclopedia of Genes and Genomes (KEGG) pathway enrichment and the gene ontology (GO) term enrichment were statistical significance (p < 0.05). According to the relations between the genes and the statistically significant biological processes as well as the relations among miR-3568, genes, and the biological processes, a miR-3568-biological processes-gene network was built.

Transcription factor analysis of the miRNA-regulated target genes in SCII

The target genes of DEmiRNAs in SCII were submitted to the TFactS database (http://www.tfacts.org/), and the transcription factors regulating the target genes of DEmiRNAs were predicted by the false discovery rate, E value, q value, and p value. To obtain reliable transcription factors, the false discovery rate, E value, q value, and p value should all be lower than 0.05. The transcription factors of the target genes of DEmiRNAs were counted, respectively. The unique and common transcription factors were compared.

Rat model

Male Sprague Dawley (SD) rats, 8 weeks, weighing 200–250 g, used in the SCII model were purchased from Liaoning Changsheng Biotechnology Co., Ltd. This study was approved by the Ethics Committee of China Medical University (CMU2020266). SCII was induced in the rats via a cross-clamped aortic arch (Li et al., 2014a, 2014b). Briefly, upon anesthesia by intraperitoneally injecting 4% sodium pentobarbital (50 mg/kg; Beyotime Biotechnology, China), endotracheal intubation (24 g trocar sleeve) and lung ventilation was accomplished with a small-animal ventilator (Harvard Apparatus, Holliston, MA, USA; tidal 15 mL/kg, breathing frequency 80–100 times/min, breathing ratio 1:1). Body temperatures were kept at 37.5 ± 0.5 °C and monitored with a rectal probe. Under aseptic conditions, the left common carotid artery was exposed in the middle of neck. Then, the aortic arch was uncovered through a cervicothoracic incision. Under direct vision, the aortic arch was cross-clamped for 14 min between the left carotid artery and the left subclavian artery to induce ischemia. A catheter (24 g trocar sleeve) was inserted into the femoral artery for blood pressure measurements. After ischemia confirmation (90% reduction of the flow assessed at the femoral artery using a laser Doppler blood flow monitor (Moor Instruments, Axminster, Devon, UK)), the clamping was removed, followed by 24-h reperfusion. This procedure was performed on the sham animals, except for blockade.

Interventions

Thirty-two male SD rats were assigned randomly to four groups: (1) sham group; (2) SCII group; (3) SCII + anti-miRNA oligonucleotides (AMOs) of miR-3568 group; and (5) SCII + NC-miR-3568 group. Rats were intrathecally injected with AMOs (Li et al., 2015b). A synthetic miR-3568 AMO (AMO-3568) (5′-CUGCUUCUGCACGGGAAGAACA-3′), and negative control were purchased from Jima Inc (China). Rats were intrathecally injected with liposome complexes of the oligonucleotides (50 mg/kg) and Lipofectamine® 2000 (Invitrogen, Carlsbad, CA, USA). Rats were injected once a day for three consecutive days before the surgical procedure.

Quantitative reverse transcription-polymerase chain reaction (qRT-PCR)

Rats were euthanized by sevoflurane overdose at 24 h after SCII in accordance with the established protocol by the Experimental Animal Center of China Medical University. Segments L4-L6 of the spinal cord were collected to extract total RNA with the Trizol reagent (Takara, Otsu, Japan). The RNA was reverse-transcribed into cDNA using a Prime-Script RT reagent Kit with gDNA Eraser (Takara, Otsu, Japan) (Jia et al., 2019). The levels of miRNA were measured using an SYBR Premix qRT-PCR (Takara, Otsu, Japan) on the Applied Biosystems 7500 Real Time PCR system (Takara) with U6 as an internal control. The primer sequences are shown in Table 1. The 2−ΔΔCt method was used to calculate the data.

Table 1 The primers used in this study.

miRNA/gene	Forward primer	Reverse primer	
miR-3568	TGTTCTTCCCGTGCAGAAG	–	
miR-144-3p	CGCGCGTACAGTATAGATGATGTA	–	
miR-34c-3p	TAATCACTAACCACACAGCCAGG	–	
miR-463	CTTGATAGACGCCAATTTGGGTAG	–	
miR-291a-3p	AAAGTGCTTCCACTTTGTGTGC	–	
miR-702-3p	TATATATGCCCACCCTTTACCCC	–	
miR-28-5p	CGAAGGAGCTCACAGTCTATTGA	–	
miR-199a-3p	CGACAGTAGTCTGCACATTGGTTA	–	
miR-352	CGCGAGAGTAGTAGGTTGCATA	–	
miR-743b-3p	CGCGAAAGACACCATACTGAATAGA	–	
miR-125b-2-3p	ACAAGTCAGGCTCTTGGGA	–	
miR-129-1-3p	CGAAGCCCTTACCCCAAAAAG	–	

Neurological evaluation

At 24 h after SCII, the hind limb functions were evaluated based on the Tarlov scores: 0 = no voluntary hind limb function, 1 = poor hind limb motor function with perceptible movement, 2 = joint motion present with no ability to stand, 3 = stands and walks, and 4 = normal hind limb function (Fang et al., 2015; Li et al., 2016a; Tarlov, 1972).

Western blotting

The expression levels of GATA6, GATA4, RBPJ, BCL-2, Bax, and cleaved caspase-3 in spinal cord tissues were measured with Western Blotting. Total proteins were extracted from the L4-L6 segments of the spinal cords with RIPA buffer (KangChen, China). The antibodies used were rabbit polyclonal anti-GATA6 (Affinity, AF5270, China), rabbit polyclonal anti-GATA4 (Affinity, AF5245, China), rabbit polyclonal anti-RBPJ (Affinity, DF7453, China), rabbit monoclonal anti-Bax (#2772; Cell Signaling Technology, Danvers, MA, USA), rabbit polyclonal anti-Bcl-2 (26593-1-AP, ProteinTech, Rosemont, IL, USA), rabbit polyclonal anti- cleaved caspase-3 (ab2302; Abcam, Cambridge, MA, USA), rabbit anti-GAPDH (Boster, A00227, China), and HRP-conjugated secondary antibodies (Beyotime, China).

Statistical analysis

SPSS 15.0 (IBM, Armonk, NY, USA) was used for statistical analyses. The results were expressed as mean ± standard deviation. Student’s t-test, one-way ANOVA followed by the Tukey’s test, or two-way repeated-measures ANOVA with the appropriate post hoc analysis were adopted to calculate the significant difference. p < 0.05 was defined as significant.

Results

The analysis process is demonstrated in Fig. 1. The key DEmiRNAs in SCII were identified first, then the target genes of the DEmiRNAs were predicted, GO and KEGG enrichment analysis and transcription factor analysis were conducted. The occurrence of DEmiRNAs in each of the datasets was calculated, and DEmiRNAs appearing in at least two datasets were identified as key DEmiRNAs. Moreover, qRT-PCR was adopted to measure the expression of the key DEmiRNAs. One of the key DEmiRNAs, miR-3568, was especially interesting; thus, the involvement of miR-3568 in SCII was preliminarily explored.

Figure 1 The analysis process of the study. SCII, spinal cord ischemia-reperfusion injury; DEmiRNAs, differentially expressed miRNAs.

Searching and identification of key DEmiRNAs in SCII

From the existing SCII miRNA expression profiling in rats, 23 independent miRNA expression datasets were obtained from PubMed, which provided the DEmiRNAs in the spinal cord tissues of SCII rats compared with the sham rats. SCII miRNA datasets were named based on the corresponding authors and year of publication for further study. Basic characteristics of SCII DEmiRNAs datasets were displayed in Table 2.

Table 2 Basic characteristics of SCII differentially expressed miRNA datasets.

Refs	Data set	Animals	SCII model	Time points	Samples	Assay/sequencing type	Validated	
Hu, Lv & Yin (2013)	HJR2013	Rat	Ligation of abdominal aorta just below the left renal artery	48 h after SCII	Spinal cord tissues	microRNA microarrays	real-time qRT-PCR	
Li et al. (2015a)	LL2015	Male SD rats (300 to 350 g)	Inserting 2 F-Fogarty balloon catheters through the left femoral artery into the proximal descending thoracic aorta.	48 h after SCII	Spinal cord tissues	qRT-PCR	–	
Li et al. (2015b)	LXQ2015	Male SD rats (200 to 250 g)	The aortic arch was cross-clamped between the left common carotid artery and left subclavian artery	24h and 48 h after SCII	Spinal cord tissues	microRNA microarrays	qRT-PCR	
Li et al. (2016b)	LJA2016	Male SD rats 280 to 300 g)	Occluding the abdominal aorta	0 h, 24 h and 48 h after SCII	Spinal cord tissues	microarray analysis	–	
Li et al. (2016a)	LXQ2016	Male SD rats (200 to 250 g)	The aortic arch was cross-clamped between the left common carotid artery and left subclavian artery	12 h and 48 h after SCII	Spinal cord tissues	microRNA microarrays	qRT-PCR	
He et al. (2016)	HF2016	Male Wistar rats(250 g)	The aortic arch was cross-clamped between the left common carotid artery and left subclavian artery	2 h after SCII	Spinal cord tissues	qRT-PCR	–	
Liu et al. (2017)	LK2017	Male Wistar rats(250 g)	Cross clamping the descending aorta just distal to the left subclavian artery	48 h after SCII	Spinal cord tissues	qRT-PCR	–	
Jin et al. (2017)	JRL2017	Male SD rats	Clamping the nontraumatic vascular clip on the abdominal aorta	20 days after SCII	Spinal cord tissues	RT-PCR	–	
Wang et al. (2018)	WY2018	Male SD rats (250 to 320 g)	Clamping the abdominal aorta with a bulldog clamp	NA	Spinal cord tissues	qRT-PCR	–	
Zhao et al. (2019)	ZLL2018	Male SD rats (approximately 250 g)	Cross clamping the descending aorta just distal to the left subclavian artery	6 h after SCII	Spinal cord tissues	qRT-PCR	–	
He et al. (2015)	HF2018	Male Wistar rats(230 to 270 g)	Cross-clamping the descending aorta just distal to the left subclavian artery	6 h, 12 h, 24 h and 48 h after SCII	Spinal cord tissues	qRT-PCR	–	
Yan et al. (2019)	YLH2018	Male Wistar rats(about 250 g)	Cross-clamping the descending aorta just distal to the left subclavian artery	6 h, 12 h, 24 h and 48 h after SCII	Spinal cord tissues	qRT-PCR	–	
Li et al. (2018a)	LXG2018	SD rats (200 to 220 g)	The aortic arch was exposed through a cervicothoracic approach and cross-clamped between the left common carotid artery and the left subclavian artery	48 h after SCII	Spinal cord tissues	qRT-PCR	–	
Li et al. (2018b)	LXQ2018	SD rats (200 to 250 g, 8 weeks)	The aortic arch was cross-clamped between the left common carotid artery and left subclavian artery	12 h, 24 h, 36 h and 48 h after SCII	Spinal cord tissues	qRT-PCR	–	
Liu et al. (2018)	LY2018	Rat	A 2 F-Fogarty balloon catheter was used to induce spinal cord ischemia through the left femoral artery into the proximal descending thoracic aorta	48 h after SCII	Spinal cord tissues	RT-PCR	–	
Qiao et al. (2018)	QY2018	Male SD rats (220 to 280 g)	Occluding the abdominal aorta	24 h after SCII	Spinal cord tissues	RT-qPCR	–	
Bao et al. (2018)	BN2018	Male SD rats (220 to 280 g)	Occluding the aortic arch	24 h after SCII	Spinal cord tissues	microRNA microarrays	qRT-PCR	
Wang et al. (2019)	WXY2019	Male SD rats (250 to 300 g)	Occluding between the left common carotid artery and left subclavian artery	0 h, 12 h, 24 h and 48 h after SCII	Spinal cord tissues	qRT-PCR	–	
Wang et al. (2020a)	WJ2020	–	Model of BSCB under hypoxia	–	Rat spinal cord microvascular endothelial cells and astrocyte	RT-PCR	–	
Chen et al. (2020b)	CFS2020	Male SD rats (200 to 250 g)	The aortic arch was cross-clamped between the left common carotid artery and left subclavian artery	6 h, 12 h, 24 h 36 h, 48 h and 72 h after SCII	Spinal cord tissues	microRNA microarrays	qRT-PCR	
Li et al. (2020c)	LR2020	Male SD rats (8 weeks)	The abdominal aorta was ligated with a 10-g bulldog clamp below the renal artery	1 h after SCII	Spinal cord tissues	RT-qPCR	–	
Fang et al. (2020)	HF2020	Male SD rats (250–260 g)	Occluding between the left carotid artery and the left subclavian artery	48 h after SCII	Spinal cord tissues	RT-qPCR	–	
Liu et al. (2020)	ZGL2020	Male SD rats (220–280 g)	The abdominal aorta was cross-clamped between the left renal artery and origin of the right renal artery	24 h and 48 h after SCII	Spinal cord tissues	microRNA microarrays	qRT-PCR	

The number of DEmiRNAs in each of the 23 SCII miRNA expression datasets was different (Fig. 2). A total of 151 DEmiRNAs were identified in the 23 SCII miRNA expression datasets. Several studies identified more DEmiRNAs than others, such as HJR2013, LXQ2015, and ZGL2020. The number of DEmiRNAs in dataset ZGL2020 was the largest (12 and 13 upregulated DEmiRNAs; 1 and 27 downregulated DEmiRNAs). Due to the differences in DEmiRNAs among the datasets, a systematic analysis of the SCII miRNA differential expression datasets was conducted to identify the key DEmiRNAs and the related potential biological functions in SCII.

Figure 2 Distribution of DEmiRNAs in 23 SCIIs.

(A) Upregulation of the proportion of DEmiRNAs in the SCII datasets. A total of 72 upregulated miRNAs were obtained. The number of DEmiRNAs supported by one dataset, two datasets, and three datasets was 51, nine (including an inconsistent expression miRNA: miR-22-3p), and 1. (B) Downregulation of the proportion of DEmiRNAs in the SCII datasets. A total of 79 downregulated miRNAs were obtained. The number of DEmiRNAs supported by one dataset, two datasets, and three datasets was 58, 9, and one.

DEmiRNAs appearing in at least two datasets were defined as key DEmiRNAs, and a total of 19 key DEmiRNAs were identified (Figs. 2 and 3). SCII induced 9 upregulated expressions, namely, miR-144-3p, miR-3568, miR-204, miR-30c, miR-200b, miR-463, miR-760-5p, miR-155-3p, and miR-34c-3p, and 10 downregulated expressions, namely, miR-125b-2-3p, miR-21-5p, miR-199a-3p, miR-352, miR-743b-3p, miR-28-5p, miR-291a-3p, miR-702-3p, miR-129-1-3p, and miR-136. Among the 19 key DEmiRNAs, the roles of five key DEmiRNAs in SCII, including miR-204, miR-30c, miR-21-5p, miR-155-3p, and miR-136, have been investigated. However, the roles of the remaining key DEmiRNAs have not been explored.

Figure 3 Expression of key DEmiRNAs following SCII.

(A) The expression of key upregulated DEmiRNAs following SCII. (B) The expression of key downregulated DEmiRNAs following SCII. n = 4 for per group. Data were analyzed with Student’s t-test. *p < 0.05, versus the sham group.

Although appeared in three independent datasets, LJA2016, ZGL2020, and HF2020, miR-22-3p showed inconsistent expression trends.

The expression of miR-760-5p has been validated via RT-PCR in HJR2013, and the expression of miR-200b has been validated via RT-PCR in LXQ2015. RT-PCR was adopted to explore the expression of the key DEmiRNAs that have not been studied or validated via RT-PCR. The results revealed that the expression levels of miR-144-3p, miR-3568, miR-34c-3p, and miR-463 increased significantly at 24 h following SCII (Fig. 3A), while the expression levels of miR-291a-3p, miR-702-3p, miR-28-5p, miR-199a-3p, miR-352, miR-743b-3p, miR-125b-2-3p, and miR-129-1-3p decreased significantly at 24 h following SCII (Fig. 3B).

KEGG and GO enrichment analysis of the target genes of DEmiRNAs in SCII

The target genes of the 19 key DEmiRNAs were predicted on miDRB. The minimum target score was set to 80, and the number of the target gene of the 19 key DEmiRNAs were obtained (Table 3).

Table 3 19 key DEmiRNAs in SCII.

Up DEmiRNAs	Number target genes in miRDB (target score ≥ 80)	Studied separately or not	The number of datasheets which supported	Down DEmiRNAs	Number target genes in miRDB (target score ≥ 80)	Studied separately or not	The number of datasheets which supported	
miR-144-3p	168	NO	2	miR-291a-3p	93	NO	2	
miR-3568	89	NO	2	miR-702-3p	39	NO	2	
miR-204	166	YES	2	miR-21-5p	66	YES	3	
miR-30c	348	YES	3	miR-28-5p	19	NO	2	
miR-34c-3p	29	NO	2	miR-199a-3p	60	NO	2	
miR-155-3p	15	YES	2	miR-352	21	NO	2	
miR-760-5p	27	NO	2	miR-743b-3p	134	NO	2	
miR-463	23	NO	2	miR-125b-2-3p	55	NO	2	
miR-200b	190	NO	2	miR-129-1-3p	39	NO	2	
				miR-136	35	YES	2	

According to the KEGG enrichment analysis of the target genes of the upregulated and downregulated DEmiRNAs, the involved pathways in the upregulated DEmiRNAs are cGMP-PKG and cAMP signaling pathway and that in the downregulated DEmiRNAs are Chemokine and MAPK signaling pathway. As shown in Figs. 4A–4B, GO enrichment analysis results indicated that target genes of the upregulated DEmiRNAs were markedly enriched in biological processes such as brain development and positive regulation of transcription from RNA polymerase II promoter. Target genes of the downregulated DEmiRNAs were mainly enriched in biological processes such as intracellular signal transduction and negative regulation of cell proliferation, as shown in Figs. 4C–4D.

Figure 4 The KEGG pathway analysis and the GO annotations for biological process of top 10 most significant enrichment terms for target genes of DEmiRNAs.

(A) KEGG pathway analysis for upregulation DEmiRNAs target genes. (B) KEGG pathway analysis for downregulation DEmiRNAs target genes. (C) The biological process of GO annotations for upregulation DEmiRNAs target genes. (D) The biological process of GO annotations for downregulation DEmiRNAs target genes.

Transcription factor analysis of the key DEmiRNAs target genes in SCII

The transcription factors corresponding to the key DEmiRNAs regulated target genes were analyzed. For the upregulated DEmiRNAs target genes, 11 transcription factor genes with 178 interactions were obtained. For the downregulated DEmiRNAs target genes, four transcription factor genes with 37 interactions were formed. Among the 11 transcription factor genes that regulated the key DEmiRNAs regulated target genes, four transcription factor genes could regulate the target genes of the upregulated or downregulated DEmiRNAs (Fig. 5A). The four transcription factors, including SP1, GLI1, GLI2, and FOXO3, had significant regulatory effects on the target genes of the key DEmiRNAs (Fig. 5B).

Figure 5 Transcription factor analysis of key DEmiRNAs target genes in SCII.

(A) The Venn Diagram of transcription factors of DEmiRNAs target genes. The purple-blue background represents the number of transcription factor analysis for upregulated DEmiRNAs target genes. The pink background represents the number of transcription factor analysis for downregulated DEmiRNAs target genes. (B) The transcription factor-DEmiRNAs target genes regulation network. The redtriangle represents the transcription factor; the purple hexagon represents upregulated DEmiRNAs target genes; the green hexagon represents downregulated DEmiRNAs target genes; the orange hexagon represents target genes of both upregulated and downregulated DEmiRNAs.

Verification of miR-3568’s involvement in SCII

For the 19 key DEmiRNAs not studied in SCII, miR-3568 was especially interesting. A previous study found that miR-3568 was upregulated in liver and serum in rats with alcoholic steatohepatitis and associated with MAPK signaling pathway (Chen et al., 2013). The expression of miR-3568 also increased in matrix vesicles (MV) compared with vascular smooth muscle cell (VSMC) in the rats with chronic kidney disease, indicating the role of miR-3568 in vascular calcification and/or MV formation (Chaturvedi et al., 2015). A recent study revealed that miR-3568 expression in simulated IRI-induced H9C2 cardiomyocytes increased in a time-dependent manner, which promotes simulated IRI-induced apoptosis in H9C2 cardiomyocytes through targeting TRIM62 (Li et al., 2020b). Although miR-3568 was upregulated after SCII in rats based on microRNA microarrays results (Chen et al., 2020b; Li et al., 2015b), the expression and potential function of miR-3568 in SCII has not been further explored.

Enriched biological processes for the target genes of miR-3568 were obtained through GO analysis. According to the relations between genes and statistically significant biological processes as well as the relations among miR-3568, genes, and biological processes, a miR-3568-biological processes-gene network was constructed. GO analysis showed that miR-3568 target genes were enriched notably in several biological processes (p < 0.05), and GATA6, GATA4, and RBPJ were enriched in several biological functions (Fig. 6).

Figure 6 A miR-3568-biological processes-gene network.

In addition, the SCII induced severe neurological deficits of lower extremities, while miR-3568 AMO improved neurological function (Fig. 7A). Cleaved caspase-3 was markedly upregulated in SCII compared with the sham group, and miR-3568 AMO reduced cleaved caspase-3 expression. Bax expression levels showed similar trends as cleaved caspase-3. Bcl-2 expression levels significantly decreased after SCII, and miR-3568 AMO increased Bcl-2 expression (Figs. 7B–7E). Moreover, the expression of GATA6, GATA4, and RBPJ decreased after SCII. Intrathecal injection with miR-3568 AMO attenuated this upregulation. The results were shown in Figs. 8A–8D.

Figure 7 Effects of AMO-3568 following SCII.

Effects of AMO-3568 following SCII. (A) Tarlov scores. n = 8 for per group. (B–E) The protein expression levels of cleaved caspase-3, Bax, and Bcl-2 were measured with Western blot assay. n = 4 for per group. Data were analyzed with the one-way ANOVA followed by the Tukey’s test. *p < 0.05, versus the sham group, #p < 0.05 versus the SCII group.

Figure 8 Effects of AMO-3568 on the protein expression levels of RPBJ (A, B), GATA6 (A, C) and GATA4 (A, D) were measured by Western blot assay.

n = 4 for per group. Data were analyzed by one-way ANOVA followed by the Tukey’s test. *p < 0.05, versus the sham group, #p < 0.05 versus the SCII group.

Discussion

To better understand the key miRNAs and the related molecular mechanism of SCII, 19 DEmiRNAs were identified in the pathogenesis of SCII. Among which, 9 DEmiRNAs were significantly upregulated: miR-144-3p, miR-3568, miR-204, miR-30c, miR-200b, miR-463, miR-760-5p, miR-155-3p, and miR-34c-3p; 10 DEmiRNAs were significantly downregulated: miR-125b-2-3p, miR-21-5p, miR-199a-3p, miR-352, miR-743b-3p, miR-28-5p, miR-291a-3p, miR-702-3p, miR-129-1-3p, and miR-136. The SCII-related functions of miRNAs such as miR-204, miR-30c, miR-21-5p, miR-155-3p, miR-136, and miR-22-3p have been investigated. Inhibition of miR-204 could promote autophagy and anti-apoptosis to mitigate SCII (Yan et al., 2019). Using wild-type (WT) and miR-155 global knockout mice, Awad et al. demonstrated that miR-155 activity accelerates the initial development of edema and the spreading of gray matter damage, and increases the rate of paralysis in a mouse model of thoraco-abdominal aortic aneurysm (Awad et al., 2018). Studies showed that abrogation of miR-30c protected PC12 cells against OGD-induced apoptosis and the inflammatory response and inhibited SCII through modulating SIRT1 (Wang et al., 2019). Functioning as a switch regulating the polarization of reactive astrocytes, miR-21 promoted synapsis formation and nerites growth after acute SCII (Su et al., 2019). Overexpression of miR-21 exerts anti-apoptosis effects on SCII via inhibiting the pro-apoptotic proteins Faslg and PDCD4 (He et al., 2016). miR-136 plays a vital role in CNS diseases and miR-136 overexpression alleviated cell apoptosis induced by SCII via targeting TIMP3 (Jin et al., 2017).

Studies on the other miRNAs are scarce but still far-reaching. Overexpression of miR-144-3p aggravated IRI-induced ischemic brain injury and promoted neurological dysfunction (Yao et al., 2020). Fang Liu et al. found that as a tumor suppressor, overexpression of miR-34c-3p caused a reduction in cell migration and invasion (Liu et al., 2015). miR-463 works as a negative regulator by targeting small proline-rich repeat protein 1A (SPRR1A) in tibial nerve regeneration (Zhao & Wu, 2019). miR-200b suppresses cell proliferation, invasion, and chemoresistance via inhibiting p70S6K1 in lung cancer (Jin et al., 2020). A recent study showed that miR-219a-3p could improve osteoblast differentiation, cellar activity, and ALP activity of BMSCs (Li et al., 2020a). As a tumor suppressor in RCC, miR-28-5p exerts multiple antitumor effects by directly inhibiting RAP1B (Wang et al., 2016). In prostate cancer cells, miR-199-3p reduced invasion and proliferation via targeting Smad1 (Qu et al., 2017). Upregulation of miR-352 resulted in autophagic lysosome dysfunction via inhibiting LAMP2 and CTSL1 (Song et al., 2018). Li et al. found that plasma exosomal miR-125b-2-3p could serve as blood-based biomarkers for diagnosing and monitoring ischemic stroke patients (Li et al., 2017). miR-129-1-3p functions as a tumor inhibitor via targeting BDKRB2 in gastric cancer (Wang, Luo & Guo, 2014). miR-760 inhibited the neuroprotective effect of NaHS against injury induced by myocardial IRI via reducing the expression of DUSP1 (Ren et al., 2020). There are few studies about miR-702-3p and miR-743b-3p.

Interestingly, miR-22-3p was downregulated in the SCII sample in the data set HF2020 but upregulated in LJA2016 and ZGL2020. The function and expression of miR-22-3p need further exploration. miR-22-3p plays a crucial role in suppressing tumors via inhibiting cellular invasion, migration, and proliferation in hepatocellular carcinoma and melanoma (Chen et al., 2016; Li, Tang & Duan, 2019). It has been reported that miR-22 acts as a potential marker in the diagnosis of astrocytoma in the thoracic spinal cord, and miR-22-3p facilitated M2 polarization of macrophages and inhibited inflammation and motor dysfunction, thus alleviating SCII (Fang et al., 2020; Ohnishi et al., 2017).

In the present study, cGMP-PKG and cAMP signaling pathways are involved in the target genes of the upregulated DEmiRNAs, while the Chemokine signaling pathway and MAPK signaling pathway are enrichment pathways of the downregulated DEmiRNAs target genes. The results are consistent with previous studies, which implicate the involvement of the above pathways in SCII (Chen et al., 2020c; Yu et al., 2018). Inhibition of reactive oxygen species reduced the MAPK pathway in the spinal cord following limb IRI in rats (Choi et al., 2015). CXCL13/CXCR5 axis promoted the development of SCII via ERK-mediated pathways (Chen et al., 2020c). During early-phase SCII, CXCL10/CXCR3 axis was related to inflammatory pain (Yu et al., 2018).

Furthermore, the key DEmiRNAs target genes are strongly regulated by transcription factors such as SP1, GLI1, GLI2, and FOXO3. SP1 was reported as a widely expressed DNA-binding protein containing a C2H2 zinc finger structure, which modulated gene transcription in various physiological and pathological processes (Wang et al., 2020b). SP1 and its family of related protein factors are implicated in various essential biological processes, such as cell growth, differentiation, carcinogenesis, and apoptosis (Vizcaíno, Mansilla & Portugal, 2015). GLI1 (Glioma-associated oncogene protein 1) family of transcription factors have three members that answer to signaling from Hedgehog and other signaling together, regulating target gene expression (Sterling et al., 2006). GLI1 can be translocated from cytoplasm to nucleus, thus activating the Hedgehog signal pathway and mediating transcription and expression of many nuclear target genes, subsequently regulating cell proliferation, apoptosis, migration, and invasion (Guo et al., 2015; Xu et al., 2014b). GLI2 played a significant role in the pathogenesis of cancer, and some studies implicated that GLI2 mediated regulation of cytokines in TME to promote cancer cell biology (Elsawa et al., 2011; Han et al., 2017). FOXOs transcription factors play crucial roles in stress resistance, inflammation, metabolism, autophagy, apoptosis, and proliferation (Zhou et al., 2019). The function of FOXO3 is highly regulated via posttranslational modifications, such as methylation, acetylation, and phosphorylation (Hedrick et al., 2012; Tia et al., 2018; Wang, Hu & Liu, 2017). Zhou et al. suggested that the activation of FOXO3 could induce brain autophagy and contribute to brain damage after IRI (Zhou et al., 2019). A recent study demonstrated that pyroptosis of cardiomyocytes in IRI was regulated by miR-149 via the directly targeting of FOXO3 (Lin et al., 2019).

Among the key DEmiRNAs not studied in SCII, miR-3568 was especially interesting. A recent study showed that miR-3568 markedly aggravated IRI-induced H9C2 cardiomyocytes apoptosis and decreased the expression of Bcl-2 and Survivin (Li et al., 2020b). This study implicated that miR-3568 might have antiapoptotic potentials. One major mechanism of SCII is neuronal apoptosis (Li et al., 2018b). In this study, SCII caused severe neurological deficits of lower extremities, while miR-3568 AMO improved neurological function. In addition, cleaved caspase-3 was markedly upregulated in SCII compared to the sham group, and AMO-3568 reduced cleaved caspase-3 expression. Bax expression levels showed a similar trend as cleaved caspase-3. Bcl-2 expression levels decreased significantly after SCII and miR-3568 AMO increased Bcl-2 expression. The expression of GATA6, GATA4, and RBPJ decreased after SCII. Intrathecal injection with miR-3568 AMO attenuated this upregulation. The mammalian GATA transcription factors comprised 6 members. Kamnasaran et al. demonstrated the GATA6 nuclear expression in endothelial cells, choroids plexus epithelium, astrocytes, and neurons (Kamnasaran & Guha, 2005). GATA6 upregulated p53 and p21 mRNA to inhibit tumorigenesis in vivo and lung cancer cell growth in vitro (Chen et al., 2020a). GATA4 has been identified as an antiapoptotic protein that protects cardiomyocytes against hypoxia, IRI, and doxorubicin-induced apoptosis (Kobayashi et al., 2006). A study has shown that GATA4 was reduced by OGD/R-induced neuronal apoptosis, indicating a neuroprotective function of GATA4 (Xiao, Kong & Hu, 2018). RBPJ is a key transcription factor downstream of receptor activation in the canonical Notch signaling pathway (Zheng et al., 2009). RBPJ-deficient pericytes induced pathogenic transformation of the vasculature resembling CCMs at the morphological and molecular level and contribute to bigger stroke lesions upon ischemic insult (Diéguez-Hurtado et al., 2019). He et al. indicated that the RBPJ-mediated Notch signaling might be involved in reducing cardiomyocyte apoptosis after myocardial infarction (He et al., 2018). Considering previous studies and the results in this study, miR-3568 might be involved in the apoptosis in SCII. GATA6, GATA4, and RBPJ might be the target genes of miR-3568 for regulating apoptosis after SCII.

In conclusion, 19 key miRNAs and the underlying molecular mechanism in the pathogenesis of SCII were explored, which could be potential intervention targets for SCII. Moreover, inhibition of miR-3568, one of 19 key miRNAs, preserved hind limb function after SCII by reducing apoptosis, possibly through regulating GATA6, GATA4, and RBPJ in SCII. miR-3568 may be a potential clinical target for inhibiting apoptosis in SCII.

Supplemental Information

Supplemental Information 1 Raw data.

Click here for additional data file.

Supplemental Information 2 Uncropped Gel Blots, Western Blots.

Click here for additional data file.

Supplemental Information 3 Checklist.

Click here for additional data file.

Additional Information and Declarations

Competing Interests

Author Contributions

Animal Ethics

Data Availability

The authors declare that they have no competing interests.

Fengshou Chen conceived and designed the experiments, analyzed the data, prepared figures and/or tables, authored or reviewed drafts of the paper, and approved the final draft.

Jie Han performed the experiments, analyzed the data, authored or reviewed drafts of the paper, and approved the final draft.

Dan Wang conceived and designed the experiments, performed the experiments, analyzed the data, prepared figures and/or tables, authored or reviewed drafts of the paper, and approved the final draft.

The following information was supplied relating to ethical approvals (i.e., approving body and any reference numbers):

The Ethics Committee of China Medical University approved this research (CMU2020266).

The following information was supplied regarding data availability:

The raw measurements are available in the Supplemental Files.

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
