# Peer review of "Identification of key microRNAs and the underlying molecular mechanism in spinal cord ischemia-reperfusion injury in rats"

_PeerJ, doi:10.7717/peerj.11454_

## Round 0.1 · original submission · Major Revisions

Please follow the requests from both expert reviewers.

Reviewer 1 ·

Basic reporting

The manuscript should be revised by a native English speaker.

Raw data was not shared.

Experimental design

The authors should inform when the Pubmed search was performed. Also, if any exclusion criteria were applied, this should be described.

Figure 4: Which rat strain was used?

Figure 4B-E: The authors should describe how the statistical analysis was performed. Which test was used? Which software was used? Were the normality of data and the homogeneity of variances assessed?

Table 2: the authors should include more information about the studies, such as the time points, the models of SCII, as well as more information about the animals (age, sex and strain).

Table 3: I suggest that the authors inform the number of times that each DEmiRNA was found in independent datasets. Moreover, considering that each miRNA was not evaluated in all datasets, it would be interesting to show the percentage of times that each DEmiRNA was found.

I also suggest to include a table showing the discrepant results (i.e., DEmiRNA that were upregulated or downregulated in at least 2 studies, but that appeared with a change of expression in the opposite direction in another dataset).

Validity of the findings

It is not completely clear why only miR-3568 was selected for the validation study. If other miRNAs were tested and there were inconclusive results, these results should be shown.

Additional comments

Figure 4b: the label of the y axis should be corrected.

Reviewer 2 ·

Basic reporting

No comment.

Experimental design

No comment.

Validity of the findings

No comment.

Additional comments

This manuscript aims to interrogate differentially expressed miRNAs in rat models of spinal cord ischemia/reperfusion. First, they surveyed public repositories of miRNA expression in the field and found 19 miRNAs that were consistently altered in ischemic/reperfused rats versus controls. They next attempted to determine molecular pathways and networks that could be deregulated by DEmiRNAs. Finally, they attempted to experimentally confirm alterations in miR-3568 and target genes. Overall, this is a timely and interesting manuscript involving bioinformatics analyses and experimental data. However, there are some issues that need to be addressed before publication.

1) Given that 19 miRNAs were found to be deregulated in rat models of ischemia/reperfusion (as per public data), I strongly suggest that the authors use their experimental model to validate more of these targets, as this would be an important contribution to the field.

2) I feel that the current discussion and interpretation of the obtained data fails to shed light on potential mechanisms/targets for spinal cord ischemia/reperfusion. For example, it is not clear enough how mir-3568 or the KEGG pathways could specifically mediate in the pathogenesis of spinal cord injury. Authors should also revise their discussion to more thoroughly convey their story.

3) Detailed statistical information are missing for all figures and should be included (e.g. statistical tests and parameters, sample size, definition of N).

4) Figure 4B-E: data should be presented as a scatter dot plot (for transparency), and the number of samples per group should be disclosed. In figure 4B, the graph label shows mir-374-5p, which I believe is a typo. Please check it.

5) Does suppression of miR-3568 rescue spinal cord injury/reperfusion-associated phenotypes?

---

## Round 0.2 · Minor Revisions

Please follow the final recommendations of both reviewers.

Reviewer 1 ·

Basic reporting

ok

Experimental design

ok

Validity of the findings

ok

Additional comments

The authors did a great job and the article improved. I only have the following comments:

1) Please include the catalog numbers (or RRID, if available) of the antibodies used for Western Blotting.

2) Full Western Blots should be shown in supplementary figures.

3) Bar graphs should ideally be replaced with graphs showing the individual data points.

Reviewer 2 ·

Basic reporting

This manuscript substantially improved after revision, with all requirements for basic reporting sufficiently addressed.

Experimental design

No comment

Validity of the findings

My final suggestion for the authors is that data on Figures 3, 7 and 8 are presented as scatter dot plots, so that readers can identify the whole datasets.

Additional comments

My comments were adequately addressed.

---

## Round 0.3 · accepted · Accept

Your revision has been examined and accepted.